# WE'LL FIX IT IN POST: IMPROVING TEXT-TO-VIDEO GENERATION WITH ZERO TRAINING

## ABSTRACT

Current text-to-video (T2V) generation models are increasingly popular due to their ability to produce coherent videos from textual prompts. However, these models often struggle to generate semantically and temporally consistent videos when dealing with longer, more complex prompts involving multiple objects or sequential events. Additionally, the high computational costs associated with training or fine-tuning make direct improvements impractical. To overcome these limitations, we introduce *NeuS-E*, a novel zero-training video refinement pipeline that leverages neuro-symbolic feedback to automatically enhance video generation, achieving superior alignment with the prompts. Our approach first derives the neuro-symbolic feedback by analyzing a formal video representation and pinpoints semantically inconsistent events, objects, and their corresponding frames with respect to the prompt. This feedback then guides targeted edits to the original video. Extensive empirical evaluations on both open-source and proprietary T2V models demonstrate that *NeuS-E* significantly enhances temporal and logical alignment across diverse prompts by almost 40%.

## 1 INTRODUCTION

Imagine generating the following complex scenario from a text-to-video (T2V) model:

*"An autonomous vehicle crosses an intersection after waiting for a pedestrian to cross."*

This involves three interdependent aspects: ❶ *semantic correctness* (the presence of objects such as the car and the pedestrian, and actions such as the pedestrian walking and the car stopping), ❷ *spatial coherence* (entities interact correctly in 3D space, for instance the car stops before the intersection when the pedestrian crosses), and ❸ *temporal consistency* (events occur in the correct order over time, for instance, the car stops when the pedestrian crosses and the car moves after the pedestrian is off the road).

Current research in T2V generation has primarily focused on the first two aspects. Existing methods improve visual quality and semantic accuracy by modifying rewards in diffusion models, tweaking attention maps, or making low-level architectural changes. Although these enhance visual fidelity, they fail to fix temporal misalignments and are often infeasible for models like Gen3 (Gen-3, 2024) and Pika (Pika Labs, 2023), which lack accessible model weights.

However, state-of-the-art T2V models fail catastrophically when prompted to generate events in a specific temporal order. Training-free methods that utilize excessive remarking to improve traditional T2V evaluation benchmarks, such as VBench (Huang et al., 2024), fail to improve temporal alignment since these metrics prioritize visual aesthetics. In contrast, *NeuS-V* (Sharan et al., 2025) introduces a neuro-symbolic method that rigorously quantifies a text-to-video model's temporal alignment with respect to the prompt. Taking inspiration from this work, we pose the following research question:

*"Can we leverage neuro-symbolic feedback to surgically refine temporally misaligned video segments through targeted edits, thereby improving text-to-video temporal alignment?"*

We introduce *NeuS-E*, a zero-training framework for targeted video refinement. Inspired by *NeuS-V*, we devise a neuro-symbolic feedback loop that disentangles atomic events and objects (termed

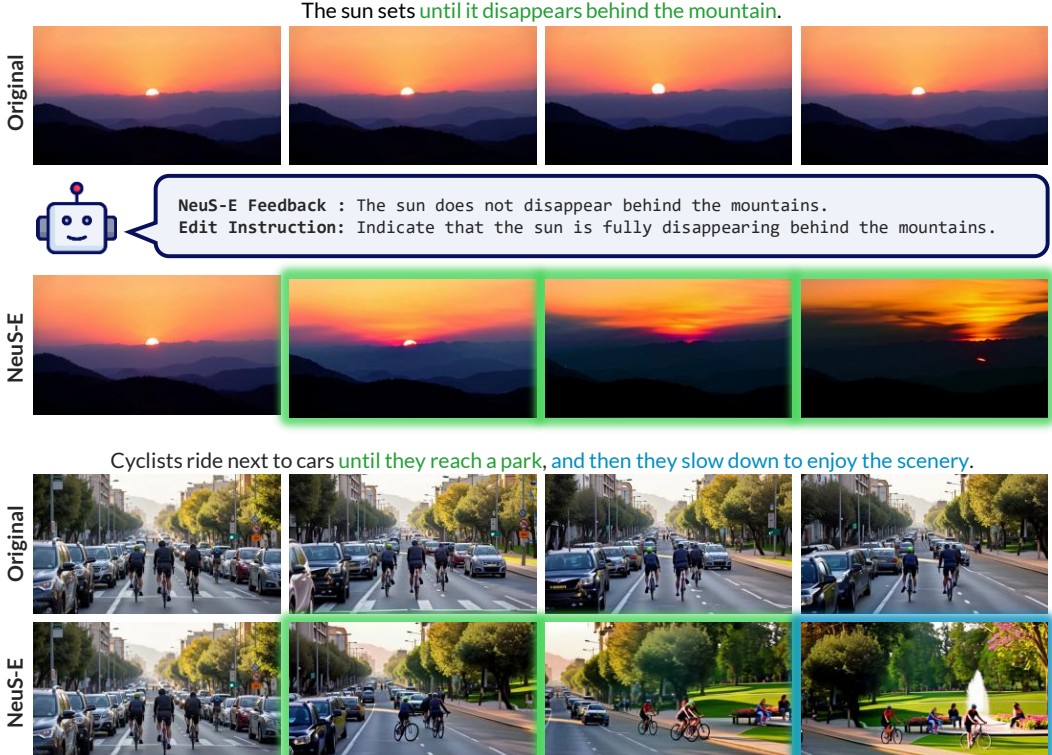

Figure 1: **NeuS-E improves the text-to-video (T2V) temporal alignment.** The border color of the frames corresponds to the identified events. Vanilla T2V models (top video) fail to generate the sunset behind the mountains. *NeuS-E* systematically identifies and surgically corrects this video segment to improve the temporal fidelity of the synthetic video with targeted feedback.

propositions) with their prompted temporal order (termed specifications). Therefore, instead of treating videos as static outputs, *NeuS-E* identifies video segments that weakly satisfy the objects and events in the prompt, edits the corresponding keyframes, and regenerates only these misaligned portions to improve temporal alignment with respect to the prompt. This iterative process produces a video that aligns with the prompt's temporal requirements without retraining the generative model. Our key contributions are summarized as follows:

- We develop a method to extract neuro-symbolic feedback, identifying weakly satisfied propositions and their corresponding problematic video segments.

- We introduce *NeuS-E*, which automatically corrects weak segments through neuro-symbolic feedback by identifying and editing the keyframe, then regenerating misaligned video segments to improve temporal alignment.

- We demonstrate that *NeuS-E* significantly boosts the temporal fidelity of both open-source (CogVideoX) and closed-source (Gen-3, Pika) models, all with zero additional training.

## 2 RELATED WORK

**Text-to-Video Generation.**    Text-to-video models such as SORA (OpenAI, 2024), GEN-3 Alpha (Research, 2024), Kling (Kuaishou, 2024), Veo (Sharma et al., 2024), Wan (Team, 2025), and PIKA (Labs, 2024) have seen significant advancements. The underlying research follows two main training strategies. The first involves end-to-end training of spatial and temporal modules using diffusion (Ho et al., 2020) or autoregressive (Hong et al., 2022; Yang et al., 2024) architectures, encompassing early latent-space works (Villegas et al., 2022; Zhang et al., 2023b; Esser et al., 2023) and numerous modern systems (Chen et al., 2023a; 2024a; Xing et al., 2024; Zhang et al., 2024b;c; Ho et al., 2022; He et al., 2023; Kong et al., 2025; Wang et al., 2024b; Team, 2025). The second strategy efficiently

adapts pre-trained image models by training only a lightweight temporal module (Blattmann et al., 2023; Guo et al., 2023; Khachatryan et al., 2023). For a broader survey, see (Cho et al., 2024). In contrast to these methods, *NeuS-E* proposes a training-free approach to video generation, for temporally complex prompts.

**Text-to-Video Refinement.** Training-free frameworks improve text-to-video generation by leveraging text-to-image (T2I) models (Meng et al., 2021; Brooks et al., 2022; Zhang et al., 2023a) to edit video frames (Jeong & Ye, 2023; Zhang et al., 2024a; Khachatryan et al., 2023; Yang et al., 2023a; Wang et al., 2024a; Goel et al., 2024b;a). Most of these methods focus on improving the temporal coherence of individual objects by rectifying cross-frame attention maps and features (Geyer et al., 2023; Jeong & Ye, 2023; Qi et al., 2023; Yang et al., 2023a; Liu et al., 2023a), by refining text prompts (Kim et al., 2024; Luo et al., 2025), by editing keyframes (Ceylan et al., 2023; Zhang et al., 2024b), or employing video-to-video diffusion models (Molad et al., 2023). While effective for object-level consistency, these approaches fail to enforce the logical order of multiple, distinct events (e.g., ensuring a person enters a car before driving away). To bridge this critical gap, *NeuS-E* uses neuro-symbolic feedback to guide edits, ensuring the final video satisfies complex temporal relationships across the entire event sequence.

**Evaluation.** Current Text-to-Video (T2V) evaluation methods include distribution-based metrics (FID, FVD, CLIPSIM) (Shin et al., 2024; Liu et al., 2023b; Jain et al., 2024; Bugliarello et al., 2023; Hu et al., 2022; Yu et al., 2023), LLM-based VQA scoring (Kou et al., 2024; Liu et al., 2024b; Zhang et al., 2024d; Li et al., 2024), and metric ensembles for visual and spatial quality like VBench, EvalCrafter, FETV, and T2V-Bench (Huang et al., 2024; Liu et al., 2024a;c; Ji et al., 2024; Feng et al., 2024; Huang et al., 2023; Chu et al., 2024). While comprehensive, these approaches primarily assess aesthetic temporal quality, such as motion and flickering (Huang et al., 2024), rather than logical event sequencing. In contrast, *NeuS-E* leverages *NeuS-V* (Sharan et al., 2025) to address this gap, using Temporal Logic (TL) to formally verify complex temporal relationships.

**Formal Verification.** Formal verification methods are used to construct symbolic representations of events and tasks across various domains. These methods have been applied in video understanding (Feichtenhofer et al., 2019; Tran et al., 2019; Medioni et al., 2001; Xu et al., 2015; Li et al., 2022; Yi et al., 2018; Chen et al., 2022), robotics (Shoukry et al., 2017; Hasanbeig et al., 2019; Kress-Gazit et al., 2009), and autonomous driving (Jha et al., 2018; Mehdipour et al., 2023), using techniques like graph-based reasoning (Yu et al., 2022; Mavroudi et al., 2020; Xiong et al., 2019), latent-space abstractions (Sarkar et al., 2015; Bertasius et al., 2021; Kroshchanka et al., 2021), or formal languages (Baier & Katoen, 2008). In contrast, *NeuS-E* leverages formal verification to provide structured feedback—identifying text-based edit instructions and keyframes—to improve Text-to-Video generation without any training.

## 3 PRELIMINARIES

We begin with a comprehensive running example of generating a promotional video using the prompt: "A person meditates by the lake and, a few seconds later, stands up for a moment before leaving" to describe the following terminologies.

**Temporal Logic.** Temporal Logic (TL) is an expressive formal language that combines logical and temporal operators to express time-dependent statements (Emerson, 1991; Manna & Pnueli, 1992). A TL specification or formula is structured around three key components: ❶ a set of atomic propositions, ❷ first-order logic operators, and ❸ temporal operators. Atomic propositions are fundamental statements that evaluate to either `True` or `False` and serve as building blocks for more complex expressions. First-order logic operators include AND ($\wedge$), OR ($\vee$), NOT ($\neg$), IMPLY ($\Rightarrow$)., and the temporal operators consist of ALWAYS ($\square$), EVENTUALLY ($\lozenge$), NEXT (X), UNTIL (U), etc. The set of atomic propositions $\mathcal{P}$, and TL specification $\Phi$ of our running example are:

$$
\begin{aligned}
\mathcal{P} &= \{\text{person is meditating, lake shore, person is standing, person is walking away}\}. \\
\Phi &= (\text{person is meditating} \wedge \text{lake shore}) \mathsf{X} (\text{person is standing} \mathsf{X} \text{person is walking away}).
\end{aligned}
\tag{1}
$$

**Video Automaton.** A video automaton formally represents a video sequence as a sequence of states and transitions (Choi et al., 2025; Sharan et al., 2025; Yang et al., 2023b). It is a form of

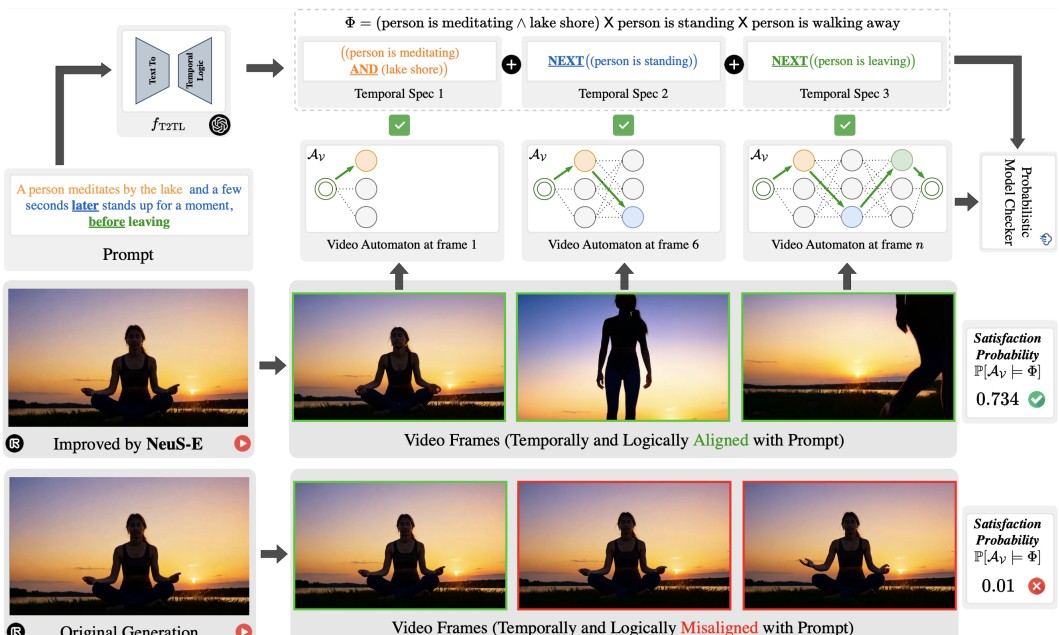

Figure 2: **Formally verify generated video with video Automaton**. The video automaton expands as new frames are added. Once fully constructed, we verify it against the TL specification. We have a very low probability of satisfaction from the initial generation, as the person neither stands nor walks away. To address this, we refine the video using *NeuS-E*, generating a better video that is temporally and logically aligned with the prompt and achieves a higher satisfaction probability.

Discrete-Time Markov Chain (DTMC) (Norris, 1998; Kemeny & Snell, 1960) used to model generated videos as each frame's transition depends only on the current frame. Since video sequences are inherently *discrete*, *finite*, and *temporal*, DTMCs can approximate frame transitions. To this end, we define the set of states in the video as $Q$ and this frame transition function as $\delta : Q \times Q \to [0, 1]$ where, given two states $q_1 \in Q$ and $q_2 \in Q$, $\delta(q_1, q_2) \in [0, 1]$ gives the probability of transitioning from $q_1$ to $q_2$. Now, the video automaton, $\mathcal{A}_\mathcal{V}$, is the tuple $\mathcal{A}_\mathcal{V} = (Q, q_0, \delta, \lambda)$ where the initial state is $q_0 \in Q$ and the label function is $\lambda : Q \to 2^{|\mathcal{P}|}$.

**Formal Verification.**  Formal verification ensures formal guarantees that the system satisfies the desired specification. (Clarke et al., 1999; Huth & Ryan, 2004). It necessitates a formal representation of the system, such as a finite-state automaton (FSA). Given the video automaton $\mathcal{A}_\mathcal{V}$, a *path* in a video is defined as a sequence of states $q_0 q_1 (q_2)^\omega$ starting from the initial state $q_0$ and $\omega$ denotes repetition. A trace corresponds to the sequence of labels $\lambda(q_0)\lambda(q_1)\lambda(q_2) \cdots \in (2^{|\mathcal{P}|})^\omega$ associated with the states along a path, where $\lambda(q)$ represents the labeling function that maps each state $q$ to a subset of atomic propositions from the set $\mathcal{P}$. The trace captures the progression of events or properties over time as observed along the path. Next, we apply probabilistic model checking (Baier & Katoen, 2008) to determine how much the *trace* starting from the initial state satisfies the temporal logic (TL) specification $\Phi$. Using these formal representations, we evaluate the videos.

**Text to Temporal Logic.**  Text prompts are converted to temporal logic conversion using LLMs (Chen et al., 2023b; Cosler et al., 2023; Mendoza et al., 2024; Yang et al., 2023b; Sharan et al., 2025) for subsequent analysis. We denote the LLM-based text-to-temporal logic (T2TL) function as $f_{\text{T2TL}} : T \to (\mathcal{P}, \Phi)$ to decompose a text prompt $T$ into a TL specification $\Phi$ and a set of propositions $\mathcal{P}$. A prompt example is presented in the Appendix.

**Neuro-symbolic Video Verification.**  We uniquely adapt a VLM to construct the video automaton and formally verify it as a symbolic method. We calibrate the VLM model to map its naive confidence to the accuracy. We present a detailed explanation of the VLM calibration process and usage in the Appendix.

**Definition 1** (Video Automaton Construction). *Given a generated video $\mathcal{V}$ and a set of atomic propositions $\mathcal{P}$, we construct a video automaton. For each proposition $p_i \in \mathcal{P}$ and frame $\mathcal{F}_n \in \mathcal{V}$, a VLM computes a semantic confidence score: $\mathcal{M}_{VLM} : p_i \times \mathcal{F}_n \rightarrow c_{i,n}$, where $c_{i,n} \in [0, 1]$ represents the confidence of $p_i$ in $\mathcal{F}_n$. For each frame $n$, we define the confidence set as: $\mathbb{C} = \{c_{i,n} \mid p_i \in \mathcal{P}, \mathcal{F}_n \in \mathcal{V}\}$. The video automaton $\mathcal{A}_\mathcal{V}$ is then generated by applying a function $\xi$ that processes the set of propositions $\mathcal{P}$ along with the confidence scores across all frames:*

$$\xi : \mathcal{P} \times \mathbb{C} \rightarrow \mathcal{A}_\mathcal{V}. \tag{2}$$

**Definition 2** (Satisfaction Probability). *Given a video automaton $\mathcal{A}_\mathcal{V}$ and a temporal logic specification $\Phi$, the satisfaction probability $\mathbb{P}[\mathcal{A}_\mathcal{V} \models \Phi]$ is computed by verifying $\mathcal{A}_\mathcal{V}$ against $\Phi$ using STORM (Hensel et al., 2020; Junges & Volk, 2021), which uses probabilistic computation tree logic (PCTL). This verification process is formalized as a probabilistic model checking function:*

$$\Psi : \mathcal{A}_\mathcal{V} \times \Phi \rightarrow \mathbb{P}[\mathcal{A}_\mathcal{V} \models \Phi]. \tag{3}$$

We provide details on the video automaton generation function in the Appendix.

## 4 METHODOLOGY

Given a synthetic video $\mathcal{V}$ generated by T2V models $\mathcal{M}_{\text{T2V}} : T, I \rightarrow \mathcal{V}$, where $T$ represents the input text prompt, and $I$ is the *optional* input image, our objective is to refine $\mathcal{V}$ to improve its temporal and logical consistency with the intended semantics of $T$. We introduce *NeuS-E*, a neuro-symbolic framework that operates through the following iterative steps:

- **Step 1: Decompose & Represent.** The prompt $T$ is decomposed into a TL specification, and a video automaton is constructed from $\mathcal{V}$ using a VLM.

- **Step 2: Identify Errors.** By analyzing the video automaton, the framework identifies weak propositions and their corresponding frames, pinpointing temporal misalignments.

- **Step 3: Refine & Iterate.** The identified errors are converted into keyframe editing instructions to guide a video editing pipeline. This process is repeated until the refined video meets a predefined coherence threshold.

### 4.1 NEURO-SYMBOLIC VIDEO VERIFICATION.

First, we generate the video with the prompt without a input image: $\mathcal{V} = \mathcal{M}_{\text{T2V}}(T, \text{None})$. Then, we decompose $T$ into the TL specification $\Phi$ and the set of propositions $\mathcal{P}$. Given $\Phi$ and $\mathcal{P}$, we construct the video automaton as described in Definition 1 and verify it according to Definition 2. Finally, we perform a comprehensive neuro-symbolic verification of the generated video. Further details are provided in Figure 2.

### 4.2 IDENTIFYING THE WEAKEST PROPOSITION.

In our running example (see Figure 2), the generated video fails to satisfy the full Temporal Logic (TL) specification—$(A \wedge B) \mathsf{X} C \mathsf{X} D$—because it omits the events corresponding to propositions C and D. To pinpoint which proposition is the weakest link, we systematically measure the impact of each one on the video's overall satisfaction probability, $\mathbb{P}[\mathcal{A}_\mathcal{V} \models \Phi]$.

Our method creates a hypothetical scenario for each proposition $\tilde{p}_i \in \mathcal{P}$ where we assume it is perfectly satisfied. Intuitively, we assess the video against partial specifications, such as $(B) \mathsf{X} C \mathsf{X} D$, $(A) \mathsf{X} C \mathsf{X} D$, $(A \wedge B) \mathsf{X} D$, and $(A \wedge B) \mathsf{X} C$, where each proposition's confidence score is systematically adjusted to find the weakest proposition. We achieve this by generating an adjusted confidence score set $\tilde{\mathbb{C}}$ that forces the confidence of the proposition under evaluation, $\tilde{p}_i$, to 1.0, while all other scores remain unchanged:

$$\tilde{\mathbb{C}} = \{\tilde{c}_{j,n} : \tilde{c}_{j,n} = 1.0 \text{ if } p_j = \tilde{p}_i; \quad \tilde{c}_{j,n} = c_{j,n} \text{ if } p_j \neq \tilde{p}_i, \forall p_j \in \mathcal{P}\}. \tag{4}$$

Using this adjusted set, we construct a new evaluating video automaton $\tilde{\mathcal{A}}_{\mathcal{V},i} = \xi(\mathcal{P}, \tilde{\mathbb{C}})$ and compute a new satisfaction probability $\mathbb{P}[\tilde{\mathcal{A}}_{\mathcal{V},i} \models \Phi] = \Psi(\tilde{\mathcal{A}}_{\mathcal{V},i}, \Phi)$. The proposition whose forced

satisfaction causes the largest increase in this probability is identified as the weakest. We formalize this by finding the proposition $p_i^*$ that maximizes the difference, $\delta_i$:

$$p_i^* = \arg\max_{i \in |\mathcal{P}|}\{\delta_i \ : \ \delta_i = \mathbb{P}[\tilde{\mathcal{A}}_{\mathcal{V},i} \models \Phi] - \mathbb{P}[\mathcal{A}_{\mathcal{V}} \models \Phi]\}. \tag{5}$$

The proposition $p_i^*$ with the highest $\delta_i$ value is the component most responsible for the video's failure to align with the prompt, allowing us to target it for refinement.

### 4.3 LOCALIZING THE INFLUENCE OF THE WEAKEST PROPOSITION.

Given the weakest proposition $p_i^*$, we now localize its influence across $\mathcal{F}_n \in \mathcal{V}$ with respect to $p_i^*$. In our running example, after we identify the weakest proposition (*e.g.,* "person is standing") from the original generation, we determine the most impacted frames by the proposition, specifically the middle frame in the Figure 2.

For each frame $\mathcal{F}_n$, we construct the evaluating automaton $\tilde{\mathcal{A}}_{\mathcal{V},n}$ for the video segment from the initial frame upto $\mathcal{F}_n$. We construct a new confidence measure $\hat{c}_{i,n} = 1.0$ for the weakest proposition $p_i^*$ at $\mathcal{F}_n$. For all other frames $\{\mathcal{F}_m \ : \ m \in [0, n-1]\}$ and propositions $p_i \in \mathcal{P}$ including $p_i^*$, we set $\hat{c}_{i,n} = c_{i,n} + \gamma$, where $\gamma$ introduces controlled noise to enhance numerical stability in the model-checking process. Subsequently, we define the set of per-frame satisfaction probabilities $\mathbb{Z} = \{\mathbb{P}[\tilde{\mathcal{A}}_{\mathcal{V},n} \models \Phi] = \Psi(\tilde{\mathcal{A}}_{\mathcal{V},n}, \Phi) \ | \ \mathcal{F}_n \in \mathcal{V}\}$. Finally, we obtain the most impacted frame $\mathcal{F}_n^*$ by $\tilde{p}_i$ as follows:

$$\mathcal{F}_n^* = \arg\max_{n \in \{1,2,...,N\}}\{z_n \in \mathbb{Z} \mid z_n = \mathbb{P}[\tilde{\mathcal{A}}_{\mathcal{V},n} \models \Phi]\}, \tag{6}$$

where $N$ is the total number of frames in $\mathcal{V}$, and $\mathcal{F}_n^*$ denotes the frame at which enforcing $p_i^*$'s certainty yields the greatest satisfaction probability, thus identifying the critical time-step most sensitive to $p_i^*$'s variability.

### 4.4 VIDEO REFINEMENT USING NEURO-SYMBOLIC FEEDBACK

We refine the video iteratively to enhance the temporal consistency of the generated video $\mathcal{V}$ through the above-mentioned neuro-symbolic feedback, which includes the baseline satisfaction probability $\mathbb{P}[\mathcal{A}_{\mathcal{V}} \models \Phi]$, the weakest proposition $p_i^*$, and the most impacted frame $\mathcal{F}_n^*$,

**Video trimming.** We index the frame $\mathcal{F}_n^*$ in video $\mathcal{V}$ and trim the video to this index to obtain a segment to $\mathcal{F}_n^*$ as: $\mathcal{V}_{\text{trimmed}} = \mathcal{V}[0 : n^*]$, where $n^*$ is the index of $\mathcal{F}_n^*$. This isolates the portion of the video most affected by $p_i^*$, preparing it for targeted regeneration.

**New Video Segment Generation.** We use an LLM to generate a new prompt $T_{\text{new}}$ to generate the next video segment after $\mathcal{V}_{\text{trimmed}}$ by providing the weak proposition $p_i^*$ along with the original prompt $T_{\text{video}}$. We present all prompts in the Appendix.

**Iterative Video Generation and Edition.** First, we merge the new video segment $\mathcal{V}_{\text{new}}$ with $\mathcal{V}_{\text{trimmed}}$, and repeat the neuro-symbolic feedback process as above to obtain the weakest proposition and the most impacted frame. This process is iterated until $\mathbb{P}[\mathcal{A}_{\mathcal{V}} \models \Phi]$ surpasses a predefined threshold or the maximum iteration limit is reached. We present the algorithm in the Appendix.

### 5 EXPERIMENTAL SETUP

Following *NeuS-V*, we use GPT-4o (OpenAI, 2023) to decompose prompts into constituent propositions and generate temporal logic specifications. InternVL2.5-8B (Chen et al., 2024b) serves as the VLM for obtaining per-frame confidence scores during automata construction. For further details on this process, we refer readers to *NeuS-V* (Sharan et al., 2025).

Once our method identifies the weakest proposition and the most impacted frame, we use GPT-4o to generate the video continuation prompt. Then, the video is regenerated using the continuation prompt. This process iterates until either (1) the *NeuS-V* score surpasses 0.7, or (2) the number of iterations matches the number of propositions—ensuring that each weak proposition undergoes at

| Prompts | | Gen-3 | | Pika-2.2 | | CogVideoX-5B | |
|---|---|---|---|---|---|---|---|
| | | Original | Edited | Original | Edited | Original | Edited |
| By Theme | Nature | 0.581 | 0.677 (+0.096) | 0.579 | 0.856 (+0.277) | 0.481 | 0.623 (+0.142) |
| | Human & Animal Activities | 0.613 | 0.767 (+0.153) | 0.638 | 0.872 (+0.235) | 0.493 | 0.596 (+0.103) |
| | Object Interactions | 0.610 | 0.721 (+0.111) | 0.420 | 0.707 (+0.287) | 0.454 | 0.610 (+0.156) |
| | Driving Data | 0.546 | 0.611 (+0.065) | 0.676 | 0.810 (+0.134) | 0.565 | 0.681 (+0.116) |
| By Complexity | Basic (1 TL op.) | 0.723 | 0.781 (+0.059) | 0.694 | 0.840 (+0.146) | 0.621 | 0.738 (+0.117) |
| | Intermediate (2 TL ops.) | 0.480 | 0.633 (+0.153) | 0.480 | 0.795 (+0.315) | 0.387 | 0.540 (+0.153) |
| | Advanced (3 TL ops.) | 0.370 | 0.527 (+0.157) | 0.373 | 0.729 (+0.356) | 0.344 | 0.449 (+0.105) |
| | Overall Score | 0.587 | 0.694 (+0.107) | 0.577 | 0.811 (+0.233) | 0.499 | 0.628 (+0.129) |

Table 1: *NeuS-V* **Score Improvements.** Comparison of original and edited *NeuS-V* scores across different themes and complexity levels for Gen3, Pika, and CogVideoX models.

least one round of refinement. Finally, we use MoviePy (Zulko et al., 2015) for video trimming and stitching to integrate the edited video clips into a final MP4 (see Figure 1).

**T2V Models.**    We evaluate our method on both closed-source and open-source T2V models to demonstrate its broad applicability. Specifically, we conduct experiments on Gen-3 (Research, 2024) by RunwayML and Pika-2.2 (Labs, 2024) by PikaArt as representatives of closed-source models, and CogVideoX-5B (Yang et al., 2024) as an open-source model. While our approach is designed to be *model-agnostic* and can be applied to *any T2V system*, we select these models due to their widespread use in the community and their ability to accept image inputs.

**Dataset and Metrics.**    We use the *NeuS-V*(Sharan et al., 2025) prompt suite, which contains temporally extended prompts that challenge T2V models. It spans four themes (Nature, Human & Animal Activities, Object Interactions, Driving Data) and three complexity levels (Basic, Intermediate, Advanced). We evaluate both original and edited videos using the *NeuS-V* score to measure temporal fidelity, and VBench(Huang et al., 2024), a widely used metric for synthetic video quality.

# 6 RESULTS

Building on our experimental setup, we now evaluate *NeuS-E* through empirical analysis. Our experiments aim to answer three key questions that highlight the necessity of our approach:

1. Does *NeuS-E* boost *alignment* without sacrificing *aesthetics* on temporally extended prompts?
2. How does iterative refinement through neuro-symbolic feedback compare to a sequential generation, where complex prompts are carefully broken down and generated sequentially?
3. Does each iteration meaningfully refine the video, progressively improving temporal fidelity?

## 6.1 IMPROVING TEXT-TO-VIDEO GENERATION WITH NEURO-SYMBOLIC FEEDBACK

As outlined earlier, we benchmark state-of-the-art T2V models, including Gen-3 and Pika as closed-source models, and CogVideoX-5B (Yang et al., 2024) as an open-source alternative.

**Benchmarking *NeuS-E* Refined Videos.**    We evaluate the before-and-after performance on the *NeuS-V* prompt suite in Table 1, measuring improvements across different themes and complexity levels. By structuring results this way, we gain deeper insights into where and how *NeuS-E* enhances temporal alignment. At a glance, *NeuS-E* consistently improves text-to-video alignment across all themes and complexities. Notably, the improvement is more pronounced for higher-complexity prompts, where regular generation with T2V models struggle the most. This suggests that while baseline models perform poorly on complex temporal relationships, our refinement method helps retain high temporal fidelity even as prompt difficulty increases. Additionally, we observe that Pika-2.2 excels at following video editing instructions, achieving an over 40% improvement in *NeuS-V* scores—outperforming other models in leveraging keyframe-based refinements.

**Do humans agree?**    To further validate the effectiveness of our approach, we conducted a fully-blind, randomized A/B human evaluation. For each prompt, annotators were shown two videos

| Strategy | *NeuS-V* | | VBench | | Length |
|---|---|---|---|---|---|
| | Original | Edited | Original | Edited | |
| Neuro-Symbolic | 0.577 | 0.811 (+0.233) | 0.789 | 0.772 (−0.017) | 14.7 sec |
| Step-by-Step | 0.577 | 0.612 (+0.035) | 0.789 | 0.784 (+0.005) | 11.2 sec |

Table 2: **Ablation Study on Refinement Strategies.** Comparison of *NeuS-V* and VBench scores for Pika-2.2 using neuro-symbolic feedback versus step-by-step prompting. Neuro-symbolic feedback is key for video refinement.

(original vs. edited) in shuffled order and asked to judge which one better aligned with the caption on a five-point scale: Strongly Agree, Agree, Neutral, Disagree, Strongly Disagree. As shown in Figure 3, our edits are preferred in 52% of trials, closely mirroring the trend observed in our benchmark evaluations. Notably, Pika-2.2 shows the largest improvement, with nearly half of its videos rated as better after refinement. Other models also exhibit consistent gains, and importantly, the number of videos rated as worse remains low across all models. A substantial portion of neutral responses typically arises from (1) videos that were already well-aligned, making edits unnecessary, or (2) cases where T2V models failed to reliably follow regeneration instructions despite multiple iterations. The latter further highlights an inherent limitation of current T2V models in generating specific complex instructions.

## 6.2 ABLATION ON REFINEMENT STRATEGY – IS NEURO-SYMBOLIC FEEDBACK KEY?

One could argue that our method is simply re-prompting a T2V model until the desired output is achieved. However, existing T2V models struggle to generate temporally coherent videos, even with repeated prompting. To determine whether our approach provides a meaningful advantage, we compare it against an alternative refinement strategy. Beyond standard re-prompting, we introduce a step-by-step generation baseline. Here, we leverage our proposition decomposition algorithm but *without* neuro-symbolic feedback. Instead of

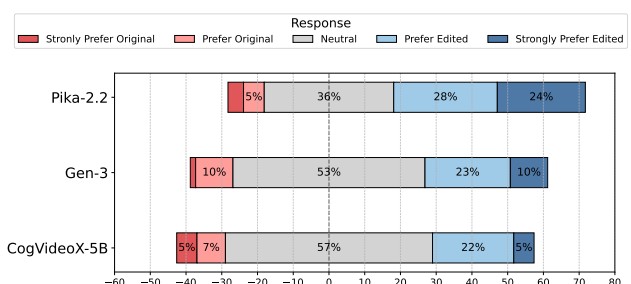

Figure 3: **Human Evaluation on Video Editing.** Diverging bar chart of human preference labels on the dataset shows that our editing pipeline improves temporal fidelity.

refining weak segments, this approach extends a video iteratively, generating each event sequentially by using the last frame of one generation as the first frame for the next.

Our results (Table 2) show that while step-by-step generation does improve temporal coherence, it falls significantly short of what neuro-symbolic feedback achieves. The key advantage of *NeuS-E* lies in its ability to extract two critical insights: (1) identifying weak propositions in the video and (2) determining the optimal time for those propositions to occur to best satisfy the temporal logic (TL) specification. Formal verification adds rigor to our approach. Additionally, we observe that step-by-step generation leads to longer videos, suggesting that without targeted neuro-symbolic feedback, the model produces redundant content rather than precisely correcting temporal inconsistencies.

## 6.3 EFFECT OF ITERATIVE REFINEMENT

In our experiments, we perform three rounds of refinement, analyzing how each iteration contributes to improving temporal alignment. Figure 4 visualizes these improvements. We observe that the violin plot widens progressively with each iteration, indicating increased variation in *NeuS-V* scores after editing. Our results show that each round of refinement provides measurable improvements, but gains plateau around the third iteration. This suggests that while early rounds effectively correct temporal inconsistencies, further refinements yield diminishing returns. Additionally, we find that CogVideoX-5B responds poorly to *NeuS-E*'s edit instructions, leading to minimal improvement compared to other models, whereas Pika-2.2 responds well.

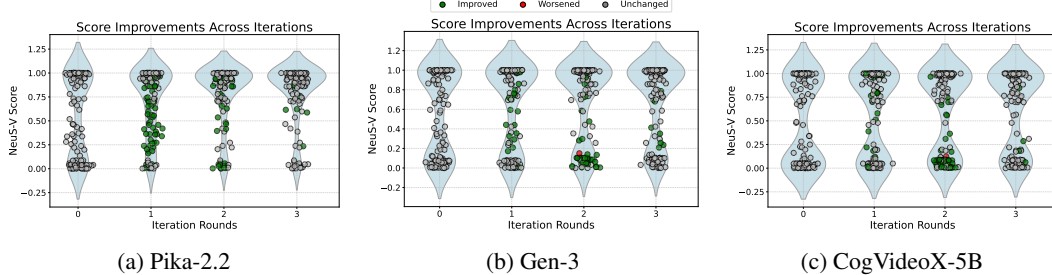

Figure 4: **Improvements from Iterative Rounds of Refinement.** Distribution of *NeuS-V* score changes with a violin plot overlay. Green/red points indicate improvements/degradation per sample.

# 7 DISCUSSION

**On Scope and Positioning.** Our work does not aim to "solve" T2V; instead, we pursue a narrow but high-impact goal: correcting the under-addressed failure mode of mis-ordered or missing events in today's T2V models, while tolerating a small, measured trade-off in aesthetic quality. Empirically, *NeuS-E* improves temporal-logic recall by +23.3 points

| Key Frame Editing | *NeuS-V* | | VBench | |
|---|---|---|---|---|
| | Original | Edited | Original | Edited |
| ✗ | 0.577 | 0.811 (+0.233) | 0.789 | 0.772 (−0.017) |
| ✓ | 0.577 | 0.835 (+0.258) | 0.789 | 0.683 (−0.106) |

Table 3: **Ablation Study on Key Frame Editing.** Comparison of *NeuS-V* and VBench scores for Pika-2.2 with and without key frame editing. Since VBench scores are highly affected by key frame modifications, this is not our default architectural choice.

with only a −1.7 point drop on VBench. Our contribution goes beyond existing methods that assess only holistic temporal coherence (Sharan et al., 2025): *NeuS-E* can pinpoint the weakest proposition in a temporal specification and localize its failure in time, revealing precisely *where* and *why* a video misaligns with the prompt. Moreover, because NeuS-E is a training-free, model-agnostic pipeline, it can be applied to *any* black-box T2V generator through targeted segment-level edits, making it broadly practical in today's landscape dominated by closed-source and proprietary video models.

**Limitations.** While *NeuS-E* consistently improves temporal fidelity, its effectiveness is constrained by the maturity of today's generation backbones. In particular, subject inconsistencies, flickering artifacts, implausible physics, and failures on abstract or highly stylized prompts expose the limits of current image and video generation models rather than our pipeline. Our ablation on keyframe editing with OmniGen (Table 3) illustrates this trade-off: while incorporating keyframe edits further boosts *NeuS-V* scores (+0.258), it also introduces a larger drop in VBench (−0.106), highlighting how *aggressive edits can destabilize visual quality*. Hence, we default to the non-keyframe-editing variant, which strikes a better balance between temporal alignment and aesthetics.

The key takeaway is that zero-training, neuro-symbolic feedback provides a principled and general mechanism for improving temporal alignment. As the underlying generation models advance, our approach will remain compatible and continue to enhance them, ensuring that temporal fidelity scales alongside improvements in visual quality.

# 8 CONCLUSION

To conclude, we propose *NeuS-E*, a training-free method to improve the generation of videos from the existing T2V model for temporally complex prompts. *NeuS-E* utilizes neuro-symbolic feedback to identify semantic discrepancies to effectively guide localized video editing. Our empirical evaluations of existing benchmarks and human evaluations demonstrate that *NeuS-E* significantly enhances temporal fidelity across diverse thematic and complexity-based categories. While *NeuS-E* attempts to bridge the gap between textual prompt complexity and video synthesis to enable temporally aligned video generation, *NeuS-E* is still limited by current T2V capabilities, and therefore, fails to meaningfully improve the video on the VBench score. We hope this work inspires further exploration into neuro-symbolic methods for advancing generative AI for long video generation.

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

## A    APPENDIX A: METHODOLOGY CLARIFICATIONS

### A.1    TEMPORAL LOGIC OPERATION EXAMPLE

Given a set of atomic propositions $\mathcal{P}$ = {Event A, Event B}, the TL specification $\Phi = \square$ Event A (read as "Always Event A") means that 'Event A' is `True` for every step in the sequence. Additionally, $\Phi = \diamondsuit$ Event B (read as "eventually event b") indicates that there exists at least one 'Event B' in the sequence. Lastly, $\Phi$ = Event A U Event B (read as "Event A Until Event B") means that 'Event A' exists until 'Event B' becomes `True`, and then 'Event B' remains `True` for all future steps.

### A.2    TEXT-TO-TEMPORAL-LOGIC

We use the prompt below to decompose a text prompt into 1) a set of propositions and 2) a TL specification.

```
fT2TL Prompt for Text to TL Specification (.md)

**System Message**:
Your input field is:
- `input_prompt` (str):  Input prompt summarizing what happened in a video.

Your output fields are:
- `input_propositions` (str):  A list of atomic propositions that correlate with
the inputted prompt formatted as [proposition_1, proposition_2, ...].
- `output_specification` (str):  The formal specification of the inputted prompt.
This is a temporal logic sequence made by combining the inputted propositions with
temporal logic symbols.
Your objective is:
- Convert the prompt into a list of propositions and a temporal logic
specification using the specified schema.
______________________________________________________________________

**User message**:
Input Prompt:  A person meditates by the lake, and a few seconds later, stands up
for a moment before leaving.

Respond with the corresponding output fields.

**Assistant message**:

Output Propositions:  ['person is meditating', 'lake shore', 'person is standing',
'person is walking away']
Output Specification:  (person is meditating ∧ lake shore) X person is standing X
person is walking away
```

**Prompt 1**: **Text to Specification Prompt.** System prompt to map prompts and propositions to the specification.

### A.3    VISION LANGUAGE MODEL

In this section, we provide the implementation details to detect the existence of propositions to label each frame in the synthetic video. We use the VLM to interpret semantics and extract confidence scores from $\mathcal{F}$ based on the text query $T$. We pass each $p_i \in \mathcal{P}$ along with the prompt to the VLM and calculate the token probability for the output response, which is either `True` or `False`. To calculate the token probability, we retrieve logits for the response tokens and compute the probability of that token after applying a softmax. Finally, the semantic confidence score is the product of these probabilities as follows:

$$c_i = \mathcal{M}_{\text{VLM}}(p_i, \mathcal{F}) \prod_{j=1}^{k} P\left(t_j \mid p_i, \mathcal{F}, t_1, \ldots, t_{j-1}\right) \ \forall p_i \in \mathcal{P}, \tag{7}$$

where $(t_1, \ldots, t_k)$ is the sequence of tokens in the response. Each term $P(t_j|\cdot) = \frac{e^{l_{j,t_j}}}{\sum_z e^{l_{j,z}}}$ is over the logits $l_{j,t_j}$ at position $j$, whereas $P(t_k|\cdot) = \frac{e^{l_{j,t_k}}}{\sum_z e^{l_{k,z}}}$ is over those at position $k$.

#### A.3.1    INFERENCE VIA VISION LANGUAGE MODELS

We use a VLM as a semantic detector. We pass each atomic proposition $p_i \in \mathcal{P}$ such as "person", "car", "person in the car", etc. Once the VLM outputs either 'Yes' or 'No', we compute the token probability of the response and use it as a confidence score for the detection.

#### A.3.2    FALSE POSITIVE THRESHOLD IDENTIFICATION

**Dataset for Calibration:**    We utilize the COCO Captions Chen et al. (2015) dataset to calibrate the following open-source vision language models – InternVL2 Series (1B, 2B, 8B) OpenGVLab (2024) and LLaMA-3.2 Vision Instruct AI (2024) – for *NeuS-V*. Given that each image-caption pair in the dataset is positive coupling, we construct a set of negative image-caption pairs by randomly pairing an image with any other caption corresponding to a different image in the dataset. Once we construct the calibration dataset, which comprises 40000 image-caption pairs, we utilize the VLM to output a 'Yes' or a 'No' for each pair.

Prompt for Semantic Detector (VLM)

```
Is there {atomic proposition (p_i)} present in the sequence of frames?
[PARSING RULE] 1.  You must only return a Yes or No, and not both, to any question
asked.
2.  You must not include any other symbols, information, text, or justification in
your answer or repeat Yes or No multiple times.
3.  For example, if the question is 'Is there a cat present in the Image?', the
answer must only be 'Yes' or 'No'.
```

**Prompt 2**: **Semantic Detector VLM.** Used to identify the atomic proposition within the frame by initiating the VLM with a single frame or a series of frames.

**Thresholding Methodology**   We can identify the optimal threshold for the VLM by treating the above problem as either a single-class or multi-class classification problem. We opt to do the latter. The process involves first compiling detections into a list of confidence scores and one-hot encoded ground truth labels. We then sweep through all available confidence scores to identify the optimal threshold. Here, we calculate the proportion of correct predictions by applying each threshold (see Figure 5). The optimal threshold is identified by maximizing accuracy, which is the ratio of the true positive and true negative predictions. Additionally, to comprehensively evaluate model behavior, we compute Receiver Operating Characteristics (ROC), as shown in Figure 5, by computing the true positive rate (TPR) and false positive rate (FPR) across all thresholds. Once we obtain the optimal threshold, we utilize it to calibrate the predictions from the VLM. We show the accuracy vs confidence plots before and after calibration in Figure 5.

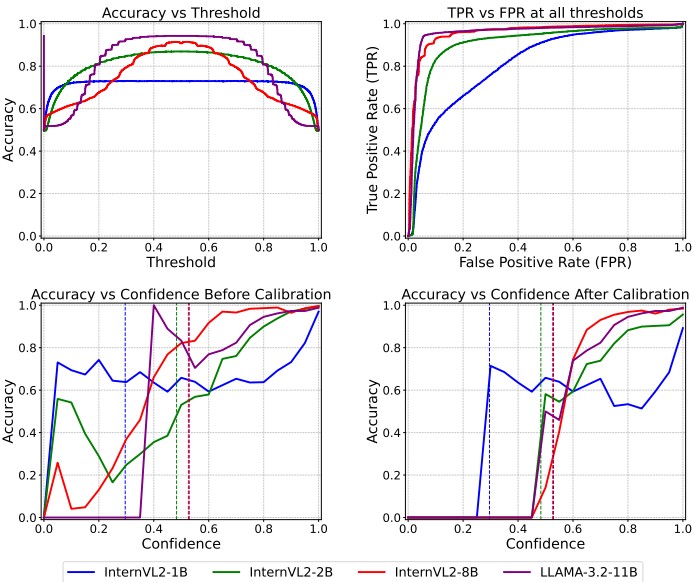

Figure 5: **Calibration Plots.** We plot the accuracy vs threshold for all VLMs on our calibration dataset constructed from the COCO Caption dataset (top left). We plot the True Positive Rate (TPR) vs False Positive Rate (FPR) across all thresholds on the top right. Finally, the bottom plots show the confidence vs accuracy of the model before and after calibration, respectively.

### A.4   VIDEO AUTOMATON GENERATION FUNCTION

Given a calibrated score set across all frames $\mathcal{F}_n$ (where $n$ is the frame index of the video) and propositions in $\mathcal{P}$, we construct the video automaton $\mathcal{A}_\mathcal{V}$ using the video automaton generation function (see Equation (2)).

$$\mathbb{C} = \{\mathbb{C}_{p_i,j} \mid p_i \in \mathcal{P}, j \in \{1, 2, \ldots, n\}\}. \tag{8}$$

As shown in Algorithm 2, we first initialize the components of the automaton, including the state set $Q$, the label set $\lambda$, and the transition probability set $\delta$, all with the initial state $q_0$. Next, we iterate over $\mathbb{C}$, incrementally constructing the video automaton by adding states and transitions for each frame. This process incorporates the proposition set and associated probabilities of all atomic propositions. We compute possible labels for each frame as binary combinations of $\mathcal{P}$ and calculate their probabilities using the $\mathbb{C}$.

### A.4.1 EDIT INSTRUCTION PROMPTS

We present different prompts to edit the keyframe and generate the new video segment in this section.

> **Prompt for Image Editing ($T_{\text{img}}$)**
>
> Add {Weak Proposition} to the image

**Prompt 3**: **Image (keyframe) Edit Instruction.** Used to edit the keyframe image to have missing semantics. The Weak Proposition will be given and passed along with the prompt

> **Prompt for New Video Generation ($T_{\text{video}}$)**
>
> You are tasked with refining video narratives generated by text-to-video models based on user feedback. For each case, you will receive two inputs:
> 1. **Original Prompt:** A description of the intended video narrative.
> 2. **Feedback:** Textual guidance on what is missing or needs adjustment in the video.

**Prompt 4**: **New Video Segment Generation Instruction.** Used to generate a new prompt to generate a new video segment.

## B ADDITIONAL RESULTS

### B.1 VBENCH SCORES

We provide the VBench scores before and after editing. We see that the edited video shows very little degradation.

| Model | Original | Edited |
|---|---|---|
| Gen-3 | 0.789 | 0.772 (−0.017) |
| Pika-2.2 | 0.799 | 0.784 (−0.015) |
| CogVideoX-5B (Yang et al., 2024) | 0.672 | 0.660 (−0.012) |

Table 4: **VBench Scores Before and After Editing.** Comparison of original and edited VBench scores across different models.

### B.2 T2VCOMPBENCH SCORES

On average, NeuS-E yields a +11% improvement across the seven T2VCompBench dimensions, driven primarily by substantial gains in Action and Interaction, which are most closely tied to correcting missing or misordered events. We include a table with category-wise improvements in Table 5.

| T2VCompBench Category | Original | Edited |
|:---:|:---:|:---:|
| Consist-Attr | 0.590 | 0.700 (+0.110) |
| Dynamic-Attr | 0.055 | 0.155 (+0.100) |
| Spatial | 0.510 | 0.600 (+0.090) |
| Motion | 0.270 | 0.390 (+0.120) |
| Action | 0.510 | 0.660 (+0.150) |
| Interaction | 0.580 | 0.710 (+0.130) |
| Numeracy | 0.240 | 0.320 (+0.080) |
| **Avg** | 0.394 | 0.505 (+0.111) |

Table 5: **T2VCompBench Scores Before and After Editing.** Category-wise improvement trends for Pika-2.2 under NeuS-E refinement.

### B.3    Human Study Annotation

We provide the annotation tool for the randomized A/B testing of the temporal alignment of the generated videos with respect to the prompt. Here, Video 1 and Video 2 are randomly assigned as the original video and the edited videos for human judgment.

## C    Use of Large Language Models

The author would like to acknowledge the use of Google's Gemini large language model in the preparation of this paper. The model served as a writing assistant, which was only used for providing valuable improvements to the clarity, conciseness, and logical flow of isolated sections, more specifically, the introduction, the methodology, and the results. It was also utilized, for assistance, with specific LaTeX formatting challenges. While the AI provided helpful refinement, the core scientific contributions, conceptual framework, and all final editorial decisions were the author's own.

## D    Algorithms

We present the algorithms for both *NeuS-E* and the video automaton generation.

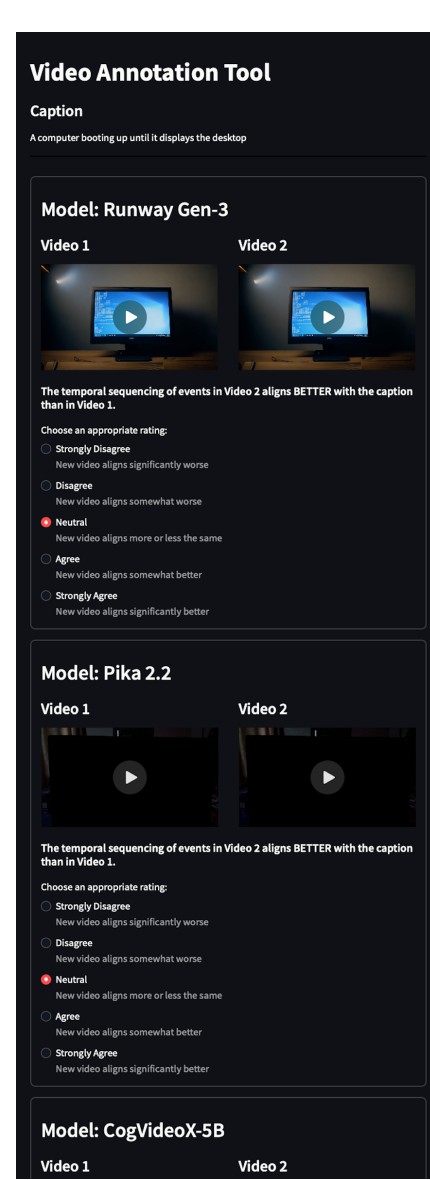

Figure 6: **Tool for Annotating Videos.** Subjects evaluate the video edited by *NeuS-E* by comparing it to its original generation across five levels: strongly disagree, disagree, neutral, agree, and strongly agree.

**Algorithm 1:** *NeuS-E*

**Require:** Text-to-video model $\mathcal{M}_{\text{T2V}}$, prompt based image to image model omnigen Text-to-TL function $f_{\text{T2TL}}(T)$, video automaton generation function $\xi(\cdot)$, probabilistic model checking function $\Psi(\cdot)$, edit instruction generation function $\text{LLM}_{\text{EIG}}$, satisfaction probability threshold $\theta$, maximum iterations $\kappa$

**Input** : Text prompt $T$, initial generated video $\mathcal{V} = \mathcal{M}_{\text{T2V}}(T)$
**Output** : Refined video $\mathcal{V}^*$

1  **begin**
2     $\mathbb{P}[\mathcal{A}_\mathcal{V} \models \Phi] \leftarrow 0$
3     $iter \leftarrow 0$
4     $\mathcal{V} \leftarrow \mathcal{M}_{\text{T2V}}(T)$                                  `// Generate a video`
5     $\mathcal{P}, \Phi \leftarrow f_{\text{T2TL}}(T)$                               `// Decompose a prompt`
6     **while** $\mathbb{P}[\mathcal{A}_\mathcal{V} \models \Phi] < \theta \wedge \text{iter} < \kappa$ **do**
7        $\mathbb{C} \leftarrow \{\}$
8        **for** $n = 0$ **to** $\text{length}(\mathcal{V})$ **do**
9           **for** $p_i \in \mathcal{P}$ **do**
10             $c_{i,n} \leftarrow \mathcal{M}_{\text{VLM}}(p_i, \mathcal{F}_n)$
11          $\mathbb{C} \leftarrow \mathbb{C} \cup \{c_{i,n}\}$
12       $\mathcal{A}_\mathcal{V} \leftarrow \xi(\mathcal{P}, \mathbb{C})$
13       $\mathbb{P}[\mathcal{A}_\mathcal{V} \models \Phi] \leftarrow \Psi(\mathcal{A}_\mathcal{V}, \Phi)$
14       **if** $\mathbb{P}[\mathcal{A}_\mathcal{V} \models \Phi] \geq \theta$ **then**
15          **Break**

                                 `// Identify the weakest proposition`
16       $\text{max\_delta} \leftarrow 0$
17       $p_i^* \leftarrow \text{None}$
18       $i^* \leftarrow \text{None}$
19       **for** $p_i \in \mathcal{P}$ **do**
20          $\tilde{\mathbb{C}} \leftarrow \mathbb{C}$
21          **for** $n = 0$ **to** $\text{length}(\mathcal{V})$ **do**
22             $\tilde{c}_{i,n} \leftarrow 1.0$
23          $\tilde{\mathcal{A}}_{\mathcal{V},i} \leftarrow \xi(\mathcal{P}, \tilde{\mathbb{C}})$
24          $\mathbb{P}[\tilde{\mathcal{A}}_{\mathcal{V},i} \models \Phi] \leftarrow \Psi(\tilde{\mathcal{A}}_{\mathcal{V},i}, \Phi)$
25          $\delta_i \leftarrow \mathbb{P}[\tilde{\mathcal{A}}_{\mathcal{V},i} \models \Phi] - \mathbb{P}[\mathcal{A}_\mathcal{V} \models \Phi]$
26          **if** $\delta_i > \text{max\_delta}$ **then**
27             $\text{max\_delta} \leftarrow \delta_i$
28             $p_i^* \leftarrow p_i$
29             $i^* \leftarrow i$

                   `// Localize the influence of the weakest proposition`
30       $\text{max\_z} \leftarrow 0$
31       $\mathcal{F}_n^* \leftarrow \text{None}$
32       $n^* \leftarrow \text{None}$
33       **for** $n = 0$ **to** $\text{length}(\mathcal{V})$ **do**
34          $\tilde{\mathbb{C}} \leftarrow \mathbb{C}$
35          $\tilde{c}_{p_i^*,n} \leftarrow 1.0$
36          $\text{evan} \leftarrow \xi(\mathcal{P}, \tilde{\mathbb{C}})$
37          $\mathbb{P}[\tilde{\mathcal{A}}_{\mathcal{V},n} \models \Phi] \leftarrow \Psi(\tilde{\mathcal{A}}_{\mathcal{V},n}, \Phi)$
38          **if** $\mathbb{P}[\tilde{\mathcal{A}}_{\mathcal{V},n} \models \Phi] > \text{max\_z}$ **then**
39             $\text{max\_z} \leftarrow \mathbb{P}[\tilde{\mathcal{A}}_{\mathcal{V},n} \models \Phi]$
40             $\mathcal{F}_n^* \leftarrow \mathcal{F}_n$
41             $n^* \leftarrow n$

                                         `// Video refinement`
42       $\mathcal{V}_{\text{trimmed}} \leftarrow \mathcal{V}[0 : n^*]$         `// Trim video up to impacted segment`
43       $T_{\text{new}} \leftarrow \text{LLM}_{\text{EIG}}(p_i^*), T_{\text{video}}$     `// Generate edit instruction for $p_i^*$`
44       $\hat{\mathcal{F}}_n^* \leftarrow \text{omnigen}(\mathcal{F}_n^*, T_{\text{img}})$     `// Edit keyframe based on the prompt`
45       $\mathcal{V}_{\text{new}} \leftarrow \mathcal{M}_{\text{T2V}}(\hat{\mathcal{F}}_n^*, T_{\text{new}})$       `// Generate new video segment`
46       $\mathcal{V} \leftarrow \mathcal{V}_{\text{trimmed}} + \mathcal{V}_{\text{new}}$   `// Merge trimmed video with new segment with text prompt and keyframe`
47       $iter \leftarrow iter + 1$

**Algorithm 2:** Video Automaton Generation

**Input:** Set of semantic score across all frames given all atomic propositions
$\{\mathbb{C} = \mathbb{C}_{p_i,j} \mid p_i \in \mathcal{P}, j \in \{1, 2, \ldots, n\}\}$, set of atomic propositions $\mathcal{P}$

**Output:** Video automaton $\mathcal{A}_\mathcal{V}$

1 **begin**

2    $Q \leftarrow \{q_0\}$      // Initialize the set of states with the initial state

3    $\lambda \leftarrow \{(q_0, \text{initial})\}$      // Initialize the set of labels with the initial label

4    $\delta \leftarrow \{\}$      // Initialize the set of state transitions

5    $Q_p \leftarrow \{q_0\}$      // Track the set of previously visited states

6    $n \leftarrow \frac{|\mathbb{C}|}{|\mathcal{P}|}$      // Calculate the total number of frames $n$

7    **for** $j \leftarrow 1$ **to** $n$ **do**

8      $Q_c \leftarrow \{\}$      // Track the set of current states

9      **for** $e_j^k \in 2^{|\mathcal{P}|}$ **do**

       // $e_j^k$: unique combination of 0s and 1s for atomic propositions in $\mathcal{P}$

10        $\lambda(q_j^k) = \{v_1, v_2, \ldots, v_i \mid v_i \in \{1, 0\}, \forall i \in \{1, 2, \ldots, |\mathcal{P}|\}\}$

11        $pr(j, k) \leftarrow 1$      // Initialize probability for the label

12        **for** $v_i \in \lambda(q_j^k)$ **do**

         // Calculate probability for $e_j^k$

13          **if** $v_i = 1$ **then**

14          $pr(j, k) \leftarrow pr(j, k) \cdot \mathbb{C}_{p_i,j}$

15          **else**

16          $pr(j, k) \leftarrow pr(j, k) \cdot (1 - \mathbb{C}_{p_i,j})$

         // Add state and define transitions if the probability is positive

17          **if** $pr(j, k) > 0$ **then**

18            $Q \leftarrow Q \cup \{q_j^k\}$

19            $Q_c \leftarrow Q_c \cup \{q_j^k\}$

20            $\lambda \leftarrow \lambda \cup \{(q_j^k, \lambda(q_j^k))\}$

21            **for** $q_{j-1} \in Q_p$ **do**

22              $\delta(q_{j-1}, q_j^k) \leftarrow pr(j, k)$

23              $\delta \leftarrow \delta \cup \{\delta(q_{j-1}, q_j^k)\}$

24            **end for**

25        **end for**

26      $Q_p \leftarrow Q_c$      // Update previous state

27      **end for**

28    **end for**

// Add terminal state

29    $Q \leftarrow Q \cup \{q_{j+1}^0\}$

30    $\lambda \leftarrow \lambda \cup \{(q_{j+1}^0, \text{terminal})\}$

31    **for** $q_{j-1} \in Q_p$ **do**

32      $\delta(q_{j-1}, q_{j+1}^0) \leftarrow 1$

33      $\delta \leftarrow \delta \cup \{\delta(q_{j-1}, q_{j+1}^0)\}$

34    **end for** // Return video automaton

35    $\mathcal{A}_\mathcal{V} \leftarrow (Q, q_0, \delta, \lambda)$

36    **return** $\mathcal{A}_\mathcal{V}$

