# OpenReview forum: "We'll Fix it in Post: Improving Text-to-Video Generation with Zero Training"
_ICLR.cc/2026/Conference — Submitted to ICLR 2026_

### Official Review · Reviewer_XsBs · 2025-10-27

**Soundness:** 3
**Presentation:** 4
**Contribution:** 3
**Rating:** 8
**Confidence:** 3

**Summary:**

This work introduces NeuS-E, a refinement technique that improves semantic and temporal consistency in video generation models. This method leverages temporal logic to measure the presence of inconsistent events and objects and use such information to regenerate the affected sets of frames of the videos, in multiple iterations.

NeuS-E is designed to be applied both to open and closed source models, without any additional training, and in benchmarks it improves temporal and logical alignment by about 40% against the baselines.

**Strengths:**

The present work presents a very original result on an important problem in video generation.

The problem identified by the authors is that video models often generate semantic and temporal inconsistencies. The approach taken in this paper has two big strengths:

* it applies results from Temporal Logic together with a video automaton, which is very novel (particularly, the way in which the weakest proposition is found by assessing each partial specification is very interesting)

* this result allows to improve on existing video models, without source code or weight access, and without any additional training, just by applying iterative edits; and it is designed to continue to be useful for future models as well

The results are also impactful, providing high improvements on a variety of metrics (e.g. +23.3 points on temporal-logic recall, and temporal and logical alignment enhancements of almost 40%). But especially, the fact that this improvement is more pronounced in more challenging settings makes this result potentially more useful in production systems.

Finally, the exposition is very clear and well written.

**Weaknesses:**

There are a few weaknesses in this paper though I don't believe any of these are major:

* A discussion of latency or cost could be useful. Plotting the quality of existing systems and NeuS-E vs. their relative costs could be insightful.

  * In diffusion models, an iso-latency inference comparison could be useful (and could be obtained by adjusting the diffusion step count, for example)

* The paper proposes a baseline of sequential generation, `where complex prompts are carefully broken down and generated sequentially`. I wonder how it would compare with a baseline in which the same process is followed, but the best of N generations at every step is picked (like beam search). The value of N could be picked such that both NeuS-E and the baseline have the same cost or latency.

* As mentioned in the paper, although the temporal-logic recall shows improvements, a quality drop is measured.

  * I wonder whether the degraded quality after the refinement iterations is just caused by lower quality frame-to-frame interpolation in the video models (when compared to just text to video generation). One way to verify whether this is the case could be to compare the quality of the refinement iterations against the same amount of iterations of regenerating the same frames but while using the original prompt.

* The model name in the annotation tool is mentioned to users. I don't think this is necessary

**Questions:**

* Can you please explain a bit (in this comment section) the role of controlled noise to enhance numerical stability in the model-checking process?

* "We can identify the optimal threshold for the VLM by treating the above problem as either a single-class or multi-class classification problem. We opt to do the latter." - any thoughts on why this is chosen, and how significant this selection is?

---

> ### Comment · Reviewer_XsBs · 2025-11-25
>
> As I mentioned in my original review, I appreciate the originality of this work but after reading the other reviewers' comments I'm concerned about some of the points raised by them. Especially insightful is f7yB's comment regarding using additional metrics. I'm waiting to see the authors' replies to those concerns, depending on this answer I may need to adjust my score to a lower value, accordingly.

---

> > ### Author Response · Authors · 2025-11-25
> > **Comment for XsBs**
> >
> > We thank you for your active participation and engagement! We will post our rebuttal today. Apologies for the delay. we have been incorporating the additional metrics requested by f7yB. You will see the additional metrics soon.

---

> ### Author Response · Authors · 2025-11-26
> **Response [1/1]**
>
> We thank you for your thoughtful and encouraging assessment. We appreciate the recognition of our method’s novelty, particularly the integration of temporal logic with video automata, and its ability to improve video models without additional training or model access. We are also grateful for the reviewer’s positive remarks on the clarity of our exposition and the strength and impact of our empirical results.
>
> **Q1. Clarification on Controlled Noise (γ = 0.01) for Model-Checking Stability**
>
> We introduce a small controlled perturbation, $\gamma = 0.01$, to the VLM-derived proposition-probability prior before model checking. This adjustment is purely for numerical stability purposes. In our DTMC-based video automaton, temporal satisfaction probabilities are computed by propagating proposition confidences across frames, and values that are extremely close to 0 or 1 can cause instability during this propagation, particularly over long sequences. Such near-degenerate probabilities can lead to underflow (where probabilities collapse to 0), artificial saturation (spurious 1.0 states), and discontinuities in weakest-trace identification. Introducing a small perturbation, $\gamma = 0.01$, ensures that no state becomes fully absorbing unless it is explicitly 1.0, thereby preventing degenerate transitions in the DTMC. This follows standard practice in probabilistic model checking, where ε-level adjustments are routinely applied to avoid zero-probability transitions that destabilize satisfaction computation. The perturbation is intentionally minimal so as not to alter the semantic meaning of the underlying proposition confidences, is applied only within the verification step (not during video regeneration), and does not bias the weakest-proposition selection. Instead, it stabilizes probability propagation across the DTMC, ensuring that weakest-trace computation remains robust and reproducible.
>
> **Q2 - Why Multi-Class Threshold Selection?**
>
> We thank you for pointing out this discrepancy. Given that the VLM is tasked with identifying multiple propositions that include objects and activities from the video, this is inherently a multi-class classification problem and cannot be treated as a single-class classification problem.

---

### Official Review · Reviewer_f7yB · 2025-11-01

**Soundness:** 2
**Presentation:** 3
**Contribution:** 2
**Rating:** 4
**Confidence:** 3

**Summary:**

The paper presents Neus-E, which bases on Neus-V to identify segments in the video that does not comply with the prompt in terms of handling complex or sequential events. The method first converts the text prompt into propositions using LLMs, and builds a video automaton to find the weakest proposition that does not align with the video. Based on the proposition, the method further identifies the video segment to be re-generated and prompts the video generation model with revised prompt from LLM to replace the original segment with better alignment.

**Strengths:**

1. The paper introduces an interesting idea of using neuro-symbolic feedback for identifying video segments that misalign to the given prompt, which is then is re-generated for better alignment. The overall motivation, and the method for identifying such segments - using temporal logic and video automatons, are very sound and convincing.

2. The method is designed to improve videos in a zero-shot manner, without fine-tuning the video for better alignment. Furthermore, the method is virtually model-agnostic asides from the part for accepting image inputs for stitching video segments.

**Weaknesses:**

1. The actual method for improving the problematic video segments however, relies on iterations of re-prompting the video generation model. While it would be reasonable for the model to better understand simple, short prompts from the atomic propositions, it would still depend on blind luck that the new segment would be better. Some study showing that the re-generated segments are much more aligned to the prompts would better support the method.

2. The evaluation mostly focuses in NeuS-V, where the methodology for identifying misaligned video segments, and the evaluation metric, shares a lot in common. If the method is basically identifying regions that bottlenecks the NeuS-V score, re-generating such segments would trivially result in better NeuS-V scores. A more thorough examination, such as evaluating with T2VCompBench[1], would be helpful.

3. The paper is lacking qualitative results, and no further results could be found in the appendix or the supplementary materials. This adds up with the concerns in the quantitative results, making it unclear how the much the proposed method can improve the original videos.

[1]Sun, Kaiyue, et al. "T2v-compbench: A comprehensive benchmark for compositional text-to-video generation." Proceedings of the Computer Vision and Pattern Recognition Conference. 2025.

**Questions:**

1. Why would the step-by-step fall so short compared to the proposed method? Considering that both method rely on the point that smaller video segments generated from propositions are expected have better alignment, the best achievable alignment for both methods looks to be the same in theory. Could this imply that the "strong" segments, opposed to the edited segments from NeuS-E, to have benefitted from the overall context of the video and should be kept?

2. How often would a re-generated segment be selected again in the next iteration? This relates to the previous question, to rule out the method from simply benefitting from having more trials within the iterations.

3. Considering that the method combines multiple models, involving LLMs and VLMs, would it be also possible to identify misaligned video segments in a more simplistic manner with modern large VLMs? For example, it would be possible to simply input the video and the prompt a LVLM and request the model to identify the misaligned frames. Demonstrating that such understanding is still difficult even with state-of-the-art LVLMs would be interesting, and would further strengthen the motivation of the work for introducing neuro-symbolic feedback.

---

> ### Author Response · Authors · 2025-11-26
> **Response [1/2]**
>
> Thank you for your thoughtful assessment and for recognizing the motivation and strengths of our paper. We appreciate your constructive questions, and below we address each in detail.
>
> **W1. Regeneration still depends on blind luck.**
> We appreciate your point regarding the stochastic nature of T2V generation. However, we clarify that our method does not rely on random chance, but rather on conditional probability. While standard T2V generation involves sampling from a vast distribution, our iterative re-prompting acts as a *constraint mechanism*. By decomposing complex descriptions into simpler, targeted atomic propositions, we explicitly emphasize the missing semantic content.
>
> The regeneration prompt is *not* a simple substring of the original. It is derived through reasoning over the failed specification, targeting and emphasizing the exact atomic proposition that was missing rather than generically restating it. This effectively narrows the generative search space, significantly increasing the probability that the model aligns with the prompt.
>
> This is empirically demonstrated in **Figure 4**, which shows a clear trend of convergence: with each iterative round of targeted re-prompting, the alignment score improves, validating that the process relies on directed refinement rather than random variation.
>
> **W2. Feedback is based on the metric used for evaluation.**
> Thank you for raising this concern. NeuS-V is only used as the primary evaluation metric because it is currently the only benchmark that measures *temporal alignment* between prompted events and generated video sequences. This focus aligns directly with the goal of NeuS-E, which is to correct misordered or missing events without loss in visual quality.
>
> However, our evaluation is not circular or trivial. NeuS-E does not optimize NeuS-V directly; rather, it identifies weak propositions and regenerates only the associated segment. If NeuS-V merely rewarded re-generation, then full re-sampling or step-by-step generation would yield similar gains which our ablation in **Table 2** shows is not the case. This demonstrates that improvements arise from targeted refinement rather than metric-induced bias.
>
> To ensure that we are not overfitting to the metric, **we already report VBench scores** (L348–349). As shown in **Table 2** these scores remain stable with only a negligible drop indicating that NeuS-E does not degrade aesthetics while improving temporal alignment.
>
> In response to your helpful suggestion, we have additionally evaluated NeuS-E using T2VCompBench. On average, NeuS-E yields a **\+11% improvement** across the seven T2VCompBench dimensions, driven primarily by substantial gains in Action and Interaction, which are most closely tied to correcting missing or misordered events. We shall include a table with category-wise improvements in T2VCompBench in our paper.
>
> **W3. Lack of qualitative results.**
> We have included additional qualitative results. Please find them in the supplementary file, with our comments [(click here)](https://openreview.net/forum?id=ifJ91JSLhq&noteId=KzVbwO7O75).
>
> **Q1. Why would the step-by-step fall so short compared to the proposed method?**
> We agree with your intuition that, in isolation, generating short video segments from subprompts is theoretically an easier task for a T2V model than generating a full video from a complex prompt.
>
> However, while step-by-step generation optimizes for *local* prompt adherence, we found it falls short on *global* coherence, which our method preserves. A key strength of generating the full video first (as in our method) is that the T2V model establishes a **unified global context** for example consistent backgrounds, lighting, and object identities across the entire timeline. In contrast, step-by-step generation treats segments in isolation or effectively as “auto-regressive extensions,” which frequently leads to semantic drift (e.g., the actor's shirt changing color, or the background shifting between steps).
>
> Furthermore, in the sequential step-by-step approach, a minor hallucination in step $t$ becomes the ground truth for step $t+1$. These errors accumulate, and would likely need Neus-E-based editing to fix the problematic segments. This approach would itself be compute intensive.
>
> By generating the full video first and then applying surgical edits, our method effectively combines the best of both worlds: we retain the “strong” segments that benefit from global context (as suggested by you) and only subject the “weak” segments to the targeted, simplified generation process, while ensuring compute efficiency.
>
> We would also like to include a point from our discussion earlier (W1) regarding the regeneration prompt not being simply a substring of the original but rather one generated through reasoning, specifically emphasizing the failing proposition. Thereby, intuitively as well as empirically we perform better than a trivial step-by-step approach.

---

> ### Author Response · Authors · 2025-11-26
> **Response [2/2]**
>
> **Q2. How often would a re-generated segment be selected again in the next iteration?**
> In NeuS-E, a regenerated segment is selected again in the next iteration only if it remains the weakest proposition after re-running the full neuro-symbolic verification on the updated video. In other words, each iteration recomputes the weakest failure point from scratch; the algorithm does not repeatedly edit the same region unless the formal model checker still identifies it as the primary bottleneck. The total number of iterations is further bounded by the number of propositions (L323), so the overall number of generations is limited.
>
> To address the concern that NeuS-E might “simply be benefitting from more trials,” we note that our ablation in **Table 2** already compares against a step-by-step baseline that also performs multiple generations. NeuS-E never performs more generations than the step-by-step baseline (often fewer) while still yielding substantially higher NeuS-V gains.
>
> This indicates that the improvements are not due merely to additional sampling, but to *where* those samples are allocated: NeuS-E uses formal feedback to concentrate generations on the *most critical*, temporally misaligned segments, rather than re-sampling the entire clip or uniformly extending it. We empirically observe that repeated refinements of the exact same segment are relatively rare and typically occur only for the most challenging prompts.
>
> **Q3. Can we use a LVLM to identify misaligned segments?**
> We agree that recent LVLMs show strong capabilities in video understanding and can often provide a high-level judgment about whether a video satisfies a prompt. However, for the purpose of correction, detecting a global failure is not enough; NeuS-E requires **precise temporal localization** of where the failure occurs in order to regenerate only the affected segment.
>
> In practice, direct LVLM querying produces coarse, sequence-level judgments (e.g., “the person never stands up”), but it does not yield the **exact frame index** or transition point needed to determine where the edit should begin.
>
> Our neuro-symbolic pipeline addresses this by constructing a video automaton and applying model checking to obtain a concrete execution trace, which identifies the weakest proposition and the precise frame at which it fails. This level of temporal resolution is necessary for targeted refinement and cannot be produced by a single forward pass of a standard LVLM.
>
> Moreover, most LVLMs process videos via sparse frame sampling (e.g., 32–128 frames for an entire clip). This introduces temporal aliasing: if the critical misalignment occurs between sampled frames, the model cannot detect the transition, let alone localize it. As a result, LVLMs may correctly classify a video as misaligned but are unable to provide the actionable, frame-specific signal required for editing.
>
> We would like to include that while a VLM can provide a coarse yes/no judgment about whether a video satisfies a prompt, using the same VLM to construct a video automaton enables formal model checking, yielding a quantified satisfaction score and an interpretable trace rather than an ungrounded verbal judgment. Neus-V’s \[1\] ablations (Sec. 6.2, Fig. 6\) also verify the same idea that formally grounded VLMs provide a more robust evaluation framework.
>
> \[1\] Sharan, S. P., et al. "Neuro-symbolic evaluation of text-to-video models using formal verification." Proceedings of the Computer Vision and Pattern Recognition Conference. 2025\.

---

> ### Author Response · Authors · 2025-11-26
> **Note regarding W1/W2/Q1**
>
> A related concern is discussed in [our response to Reviewer 3](https://openreview.net/forum?id=ifJ91JSLhq&noteId=qnnZYwcCUr) (W1 Part 1), where we provide additional detail on the distinction between targeted temporal localization and generic re-prompting. We would like to kindly refer you there for complementary clarification.

---

> ### Author Response · Authors · 2025-12-02
> **T2V-CompBench Result is posted in the updated paper**
>
> We want to note that the full T2V-CompBench results are provided on page 20, covering all seven evaluation dimensions. Across these dimensions, *NeuS-E* achieves an overall improvement of approximately **+11%**. We hope this addresses your concern about the evaluation. If you are able to share any thoughts on these evaluation results, it would be greatly helpful for the new Area Chair who will conduct the final review of our submission.
>
> Regards,
>
> Authors

---

### Official Review · Reviewer_UHE3 · 2025-11-01

**Soundness:** 3
**Presentation:** 3
**Contribution:** 3
**Rating:** 4
**Confidence:** 3

**Summary:**

This paper proposes a training-free video refinement pipeline that uses neuro-symbolic feedback to automatically enhance text-to-video generation by identifying temporally misaligned segments and editing to improve prompt alignment. The method works by decomposing text prompts into temporal logic specifications, constructing video automatons to identify weak propositions and their corresponding problematic frames, then iteratively regenerating only the misaligned portions until temporal consistency is achieved.

**Strengths:**

- The conceptual aspect of the proposed approach is thoroughly developed and impressive. Using neuro-symbolic feedback as a form of structured feedback for video improvement is intuitive and makes sense.
- Extensive experiments have been conducted corresponding to the key points of the proposed pipeline.
- Consistent performance improvements have been reported with NeuS-V score.

**Weaknesses:**

While the conceptual approach is interesting, there are worries about practical performance improvements given that video generative models are primarily considered black boxes, and the absence of enough qualitative results.

- The need for the proposed pipeline fundamentally arises because video generation models fail to properly understand and reflect user prompts. However, the main element improved by the proposed NeuS-E feedback is also text prompts. Although text-based image editing models are used via keyframes as intermediaries, the origin of performance improvement could be because off-the-shelf image models simply follow prompts better.
- While quantitative analysis is provided, the amount of qualitative results provided in the paper is extremely limited and video results were not provided, which makes the practical performance of the proposed method less convincing.

**Questions:**

The biggest concern is the absence of sufficient qualitative results. How are the qualitative results? It would be good if they were provided in the appendix or supplementary materials.

---

> ### Author Response · Authors · 2025-11-26
> **Response [1/1]**
>
> We appreciate your positive evaluation of the framework and its motivation, and we welcome the questions regarding practical behavior and qualitative validation. We respond to each concern in detail below.
>
> **W1. (Part 1\) The method relies on text prompts to fix failures caused by text prompting.**
> We appreciate this observation. While NeuS-E ultimately triggers regeneration through a text prompt, the principal contribution of the pipeline is **not** prompt engineering, but *formal identification and localization* of the specific temporal failure. The challenge we address goes beyond interpreting long prompts and centers on satisfying the **correct ordering and occurrence of specific events** within the video.
>
> A naïve re-prompt (or a stronger prompt) does not address this, because it lacks a mechanism to determine *which* event failed and *where* in the timeline the failure occurred. In contrast, NeuS-E leverages neuro-symbolic verification to:
>
> 1. **Detect the weakest proposition** via temporal logic,
> 2. **Localize the failure** to an exact frame index, and
> 3. **Regenerate only that segment**, while preserving all correctly aligned portions.
>
> The regeneration prompt is therefore *not* the core improvement; it is simply the final execution step after a formally grounded diagnostic process. In other words, NeuS-E does not “improve prompts,” but **derives an actionable, localized correction signal** that standard prompting does not provide.
>
> Empirically, this distinction is reflected in our ablation (Table 2), where step-by-step prompting despite having multiple opportunities to “improve the prompt” yields far smaller gains than NeuS-E. This demonstrates that the benefit arises from *targeted localization*, not from prompting alone.
>
> We would like to note that we discuss similar ideas in detail with [our response to Reviewer 3](https://openreview.net/forum?id=ifJ91JSLhq&noteId=e9UQ8PsYEm) (W1, W2, Q1), and we kindly refer you there for additional information if necessary. But please do not hesitate to reach out if you have any questions. We would very gladly welcome your follow-up and feedback on this regard.
>
> **W1. (Part 2\) The origin of improvement could be from T2I keyframe editing models.**
> We would like to clarify that T2I editing is an architectural ablation, not the default (L466-470). All tables and plots with the exception of “Table 3: Ablation Study on Key Frame Editing” do not use T2I models. Our framework operates directly with the T2V model for iteratively refining the video segments. Since our main method achieves significant alignment improvements using the original T2V backbone, the gains cannot be attributed to external T2I models, but rather to the efficacy of our method.
>
> **W2+Q1. Lack of qualitative results.**
> We have included additional qualitative results. Please find them in the supplementary file, with our comments [(click here)](https://openreview.net/forum?id=ifJ91JSLhq&noteId=KzVbwO7O75).

---

### Official Review · Reviewer_FLya · 2025-11-01

**Soundness:** 2
**Presentation:** 2
**Contribution:** 2
**Rating:** 4
**Confidence:** 4

**Summary:**

The author introduces a zero-training video refinement pipeline that leverages neuro-symbolic feedback to automatically enhance video generation.

**Strengths:**

The author introduces an interesting zero-training video refinement pipeline that leverages neuro-symbolic feedback to automatically enhance video generation.

**Weaknesses:**

1.The author uses video editing to correct incorrect video generation. However, video editing is a more difficult task than video generation, with lower accuracy and higher computational resource requirements. Therefore, I think this approach doesn't make sense in practice. Relying on video editing for correction is less effective than improving the success rate of video generation in the first place.


2. This pipeline is too idealistic and complex. If the video generation involves complex object changes, this pipeline is likely to fail.

**Questions:**

see weakness

---

> ### Author Response · Authors · 2025-11-26
> **Response [1/1]**
>
> Thank you for taking the time to review our paper. We appreciate your identification of our paper’s main idea, leveraging neuro-symbolic feedback for zero-training temporal refinement in text-to-video generation.
>
> **W1. Video editing is more complex than video generation.**
>
> While we agree that "video editing" is difficult and impractical when it refers to full video-to-video pipelines that must process and modify every frame in a sequence, our method avoids this computational burden. By identifying a single weak proposition and one key frame (Sec. 4.2–4.4), we regenerate *only* the short segment following that frame. This makes our method considerably cheaper and more practical than full-sequence re-generation, prompt-level resampling, or the manual iterative rejection sampling currently employed by most users. Additionally, we first highlight the limitations of the standard alternative: step-by-step generation or "just regenerating the whole clip." We demonstrate and highlight the weakness of such baselines in *Table 2,*  and we show that these are significantly less effective at achieving alignment, despite requiring substantially more computation. Furthermore, as outlined in the response to Reviewer f7yB’s Question 1, sequential step-by-step generation would accumulate the errors generated per segment. Our approach therefore, provides a more practical solution to improving text-to-video temporal alignment.
>
> **W2. Concern About Idealism and Complex-Prompt Robustness.**
>
> We respectfully disagree with the characterization of the approach as “too idealistic.” Our method is not an aspirational or open-ended heuristic, but a principled, formal refinement pipeline grounded in temporal logic. This formalism directly addresses the concern regarding the stochastic nature of T2V generation: rather than relying on random sampling from a vast distribution, our method operates on **conditional probability**.
>
> By decomposing complex descriptions into targeted atomic propositions, we explicitly emphasize missing semantic content, acting as a constraint mechanism that effectively narrows the generative search space. This is empirically demonstrated in *Figure 4*, which shows a clear trend of convergence; with each iterative round of targeted re-prompting, the alignment score improves, validating that the process relies on **directed refinement** rather than random variation.
>
> Consequently, while we agree that today’s T2V models struggle with complex global transformations zero-shot, our method bypasses this limitation by applying **surgical, segment-level edits**. By localizing failures and correcting them one at a time until they pass the required alignment threshold (as defined by the automata), the design is inherently practical: it leverages the model’s strengths on small, local changes while avoiding the unpredictability of global regeneration.

---

### Author Response · Authors · 2025-11-26
**Additional qualitative results**

We have added additional qualitative results in response to reviewer requests. We would have liked to include more, **but the supplementary file must remain under 100 MB.** We will upload additional videos to our project webpage after the review period.

How to access the supplementary material

Step 1 — Download the supplementary file directly from the OpenReview page, or use this [link](https://openreview.net/attachment?id=ifJ91JSLhq&name=supplementary_material).

Step 2 — Extract (unzip) the contents of the downloaded file.

Step 3 — Open `main.html` to browse the results. If you encounter any issues, please refer to the included README files. As an alternative, you can manually view the videos inside the video folder.

---

### Author Response · Authors · 2025-12-02
**Can you give us your final comment on our rebuttal?**

Dear Reviewers,

We sincerely appreciate your insightful feedback. It is unfortunate that the review process does not allow further engagement, but we want to express that we have made every effort to address your comments and concerns in our rebuttal. Although scores cannot be updated at this stage, we would be grateful for any final remarks you may wish to share on our response. This will also help us think about our future work.

Thank you for your time and consideration. We wish you a warm and restful end of the year.

Warm regards,

The Authors

---

### Author Response · Authors · 2025-12-02
**Summary of Our Rebuttal**

We sincerely thank the reviewers and Area Chair for their thoughtful evaluation. We are encouraged by the consistent recognition across reviews that our work introduces a novel and well-motivated approach to **improving temporal alignment in text-to-video generation**.

Reviewers highlighted the originality and soundness of leveraging neuro-symbolic feedback via temporal logic and video automata (R1–R4), praised the clarity of the exposition (R2, R4), and emphasized the practicality and model-agnostic nature of our zero-training refinement pipeline (R3, R4). They also acknowledged the strength of our empirical results, including consistent improvements in NeuS-V (R2) and substantial gains such as **\+23.3 points** in temporal-logic recall and nearly **40% improvements** in temporal and logical consistency (R4), supported by extensive experiments.

During the rebuttal phase, we focused on addressing the following major concerns:

* **(R1) Practicality of “video editing” and computational cost.** We clarified that NeuS-E does **not** perform full video editing; it regenerates *only a short localized segment* identified via the weakest proposition. This makes it substantially cheaper than whole-clip regeneration or step-by-step baselines, which we showed in Table 2 to be both more expensive and less effective.

* **(R1) Pipeline is “idealistic” and may fail on complex prompts.** We explained that NeuS-E is a *principled, formal refinement pipeline*, not a heuristic: temporal logic decomposition narrows the generative search space and yields *directed, non-random improvement*, as shown by monotonic gains in Figure 4\. By applying surgical, segment-level corrections instead of global edits, NeuS-E remains practical and robust even on complex prompts.

* **(R2) Improvements may stem from prompting or keyframe T2I editing rather than neuro-symbolic refinement.** We clarified that NeuS-E’s core contribution is formal localization of the weakest temporal failure, *not* prompt rewriting. The method identifies *which* event failed and *where* it failed—something naive re-prompting cannot do—and regenerates only the affected segment. We also emphasized that T2I keyframe editing is an *ablation only* and all main results use the original T2V model, establishing that improvements are not due to external image models.

* **(R2/R3) Insufficient qualitative results.** We addressed this by adding substantial qualitative examples and commentary in the supplementary material, including side-by-side comparisons and full edited videos.

* **(R3) Regeneration may rely on random chance (“blind luck”).**  We clarified that the regeneration prompt is *not a simple substring of the original*, but a reasoned, purpose-built prompt derived from the failed temporal specification and constructed to explicitly emphasize the missing atomic proposition. Combined with precise frame localization from temporal-logic verification, this converts re-generation into targeted refinement rather than stochastic retrying.

* **(R3) Evaluation may be biased because NeuS-V is used both for supervision and measurement.** We addressed this by explaining that NeuS-E does *not* optimize NeuS-V directly—it uses only the weakest-proposition trace for localization, not for score-based optimization. We also pointed out that VBench is *already included* in our evaluation and its scores remain stable after editing. Finally, following the reviewer’s suggestion, we added **T2VCompBench** results, where NeuS-E achieves an average **\+11% improvement**, especially in Action and Interaction categories.

We believe the revisions directly address the concerns raised, and we remain confident that NeuS-E offers a principled, practical, and model-agnostic approach to improving temporal alignment in text-to-video generation. We will incorporate highlights of these discussions in the camera-ready version of our manuscript.

Given the limited time in the discussion period, we are very happy to continue the conversation and address any remaining questions.

---

### Meta-Review · Area_Chair_fXSV · 2026-01-05

**Summary:**

This paper proposes a zero-training post-processing pipeline for text-to-video generation: generate a full video, formalize the prompt as temporal-logic constraints, use a video model checking signal to identify the weakest proposition and the most impacted segment, then locally re-generate and replace the failing portion iteratively.  While the direction is interesting, I recommend rejection for three reasons. (1) Causal attribution is unclear: as raised by f7yB and UHE3, the gains may largely come from prompt rewriting regeneration rather than the neuro-symbolic refinement signal itself, and the paper does not isolate this sufficiently.
(2) Evaluation validity remains a concern: f7yB pointed out potential circularity when the same validation machinery both guides refinement and serves as the primary metric; although the authors add an external benchmark (e.g., T2VCompBench), the overall evidence is still not fully convincing for the headline claims.
(3) Practicality and evidence: concerns remain about runtime/latency and the limited qualitative evidence for complex prompts; even the supportive reviewer XsBs asked for stronger iso-latency comparisons and expressed caution pending additional metric evidence.

**Reviewer Concerns:**

Addressed by rebuttal:
Added external benchmark results (e.g., T2VCompBench) in response to metric-circularity concerns.
Added more qualitative results in the supplement (within size limits).
Clarified technical details such as small controlled noise for numerical stability in verification.
Still outstanding:
Whether improvements come from targeted neuro-symbolic feedback vs. “smarter retries and prompt edits” (f7yB, UHE3).
Remaining concerns about evaluation robustness and fair cost or latency comparisons (f7yB, XsBs).

**Reviewer Scores:**

XsBs (initial 8): likely slightly down to 6 due to lingering concerns around evaluation and iso-latency comparisons.
f7yB (initial 4): likely stays 4; external benchmark additions help, but mechanism isolation and non-circular evaluation remain core concerns.
UHE3 (initial 4): likely stays 4; skepticism that gains are driven by prompt rewriting editing rather than the proposed refinement signal is not fully resolved.
FLya (initial 4): likely stays 4; concerns are broad/practical and not clearly shifted by the rebuttal.

---

### Decision · Program_Chairs · 2026-01-26

Reject